# LEVERAGING IMAGE REPRESENTATIONS FOR BOUNDED ADVERSARIAL ATTACKS AND ROBUSTNESS

## ABSTRACT

Both classical and learned image transformations such as the discrete wavelet transforms (DWTs) and flow-based generative models provide semantically meaningful representations of images. In this paper, we propose a general method for robustness exploiting the expressiveness of image representations by targeting substantially low-dimensional subspaces inside the $L^\infty$ box. Experiments with DCT, DWTs and Glow produce adversarial examples that are significantly more similar to the original than those found considering the full $L^\infty$ box. Further, through adversarial training we show that robustness under the introduced constraints transfers better to robustness against a broad class of common image perturbations compared to the standard $L^\infty$ box, without a major sacrifice of natural accuracy.

## 1 INTRODUCTION

The deployment of deep neural networks for image classification in critical decision-making processes has raised concerns about their robustness. Despite often stellar test set accuracies, these models have also shown to be brittle in various ways in which the human vision is not. For example, a network can be fooled by suitably designed malicious perturbations that look non-suspicious or even undetectable by a human. Further, networks are also not robust when faced by real-world image corruptions such as images taken under different weather conditions.

**Adversarial robustness.** Given a neural network that makes accurate predictions on clean data, adversarial attacks (Biggio et al., 2013; Szegedy et al., 2014; Papernot et al., 2016a) compute a suitable choice of additive noise to produce erroneous predictions. For images, the noise is typically measured and bounded by an $L^p$ norm. The seminal method of projected gradient descent (PGD) (Madry et al., 2018) is a prominent example. Learning-based methods also have been used to build adversarial attacks either by leaning an embedding space using neural networks (Huang & Zhang, 2020; Baluja & Fischer, 2018), using latent space of generative adversarial networks (GAN) (Xiao et al., 2018; Wang & Yu, 2019) or using flow-based models to attack in black-box settings (Dolatabadi et al., 2020). Many approaches have been proposed to detect if an input is adversarial (Xu et al., 2018; Ma et al., 2018; Feinman et al., 2017; Metzen et al., 2017) and defend against it (Gu & Rigazio, 2014; Papernot et al., 2016b; Liao et al., 2018; Xie et al., 2019; Zhou et al., 2021). However, most of these defenses can again be broken by suitable adaptive attacks (Tramèr et al., 2020; Carlini & Wagner, 2017).

More importantly, adversarial attacks (assuming they are fast enough) can be used for adversarial training to increase robustness by first generating adversarial examples from clean training data, and then either performing standard training on these (Madry et al., 2018) or combining them with clean data to define a loss that, when minimized, better preserves the natural accuracy (Kannan et al., 2018; Zhang et al., 2019) or other variants such as (Chen et al., 2021; Rebuffi et al., 2021; Jiang et al., 2023). Further, adversarial training can be used to obtain provably robust models (Salman et al., 2019; Müller et al., 2022). However, typically the price of adversarial training is a significant drop of the classification accuracy on the unperturbed, clean data. On the other hand, provable adversarial robustness can be provided through randomized smoothing (Salman et al., 2019; Carlini et al., 2022), a sampling-based approach that scales to large models regardless of their complexity.

**Corruption robustness.** Arguably more important for practical applications, and a longstanding goal in neural network design, is robustness against distribution shifts between training data and

application data (Mintun et al., 2021; Pan & Yang, 2010; Farahani et al., 2020). One class of such shifts, and the one considered in this paper, are image corruptions. Examples include digital effects such as compression or weather conditions such as fog. Other forms of distribution shift are studied by applying abstract changes in structure and style (Hendrycks et al., 2021) to images or by sampling new versions of datasets (Recht et al., 2019).

Training networks to be robust against common image corruptions has become an active research topic, especially after the introduction of dedicated benchmarks such as ImageNet-C (Hendrycks & Dietterich, 2019). Approaches include again suitable data augmentation (Geirhos et al., 2018; Erichson et al., 2022; Zhang et al., 2017; Hendrycks et al., 2019; Park et al., 2022; Yin et al., 2022; Liang et al., 2023) in training or the use of transformed image representations. For example, training techniques relying on the discrete cosine transform (DCT) are found effective to generalize to unseen image distortions, for example, by extending the dropout technique to DCT coefficients as a form of regularization (Hossain et al., 2019). Similarly, (Duan et al., 2021) defined constraints in the DCT domain to generate adversarial examples that are less affected by JPEG compression than those obtained by pixel-based attacks.

Interestingly, (Hendrycks & Dietterich, 2019; Ford et al., 2019; Xie et al., 2020; Kang et al., 2019; Kireev et al., 2022) found that adversarial training using $L^p$ norms (without any further constraints) also yields good accuracy to several common perturbations such as blur and weather.

**Our contribution.** In this paper we offer progress in the quest for corruption robustness by presenting a powerful novel adversarial attack and associated adversarial training. We will demonstrate that the latter can yield networks with both only a small drop in accuracy on unperturbed test data and better robustness across common categories of corruptions. The key idea is to perform an adversarial attack in a meaningful subspace of a transformed image representation (e.g., the details in a wavelet-transformed image) while, at the same time, obeying the $L^\infty$ box in the image domain, i.e., staying close in pixels. In other words, our attacks operate exclusively in a meaningful subspace (as defined by the chosen transform) of the $L^\infty$ box.

Doing so is not possible with prior attacks such as PGD since the needed projections are not available in closed form for such a complex perturbation space. Instead, we use the *barrier method* from non-linear programming (Chachuat, 2007; Nocedal & Wright, 2006) to compute perturbations while satisfying the constraints without the need for projections. Doing so makes our approach efficient enough for attacks and to be integrated in adversarial training and for a wide class of image transformations including classical linear ones such as the discrete wavelet transform (DWT) or non-linear learned transforms such as flow-based models. Specifically, we contribute:

- A novel white-box attack that efficiently computes adversarial perturbations in a predefined transformed representation subspace while obeying the $L^\infty$ pixel constraint at the same time.
- Instantiations of our approach for the two classical linear transforms DCT and DWT and the nonlinear learned flow-based model Glow (Kingma & Dhariwal, 2018).
- An evaluation of our attacks against prior work on ImageNet and CIFAR-10. In particular, given the same $L^\infty$ box, we show that adversarial images found by our approach present significantly higher similarity to the originals, as verified by the learned perceptual image patch similarity metric (LPIPS) (Zhang et al., 2018).
- We show that using our attacks for adversarial training can yield excellent robustness on the image corruption benchmark CIFAR-10-C, up to on average 12.17% more accurate, with only a small drop in natural accuracy, than those trained under the full $L^\infty$ box.

## 2 ADVERSARIAL ATTACK

In this section we explain our adversarial attack based on perturbing an image in a transformed representation, while, at the same time, obeying the classical box constraint in the pixel domain. The approach is depicted as cartoon for two-dimensional images in Fig. 1 and leverages the barrier method (Chachuat, 2007; Nocedal & Wright, 2006) from non-linear programming for the occurring optimization problem.

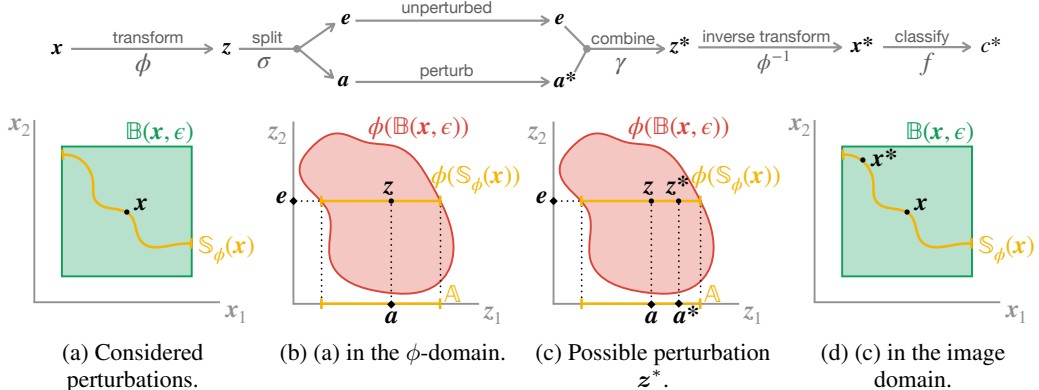

(a) Considered perturbations.

(b) (a) in the $\phi$-domain.

(c) Possible perturbation $\boldsymbol{z}^*$.

(d) (c) in the image domain.

Figure 1: High-level depiction (in two dimensions) of our approach for finding an adversarial example of an image $\boldsymbol{x}$ using a chosen transform $\phi$ and split operator $\sigma$. (a) shows the considered perturbations $\mathbb{S}_\phi(\boldsymbol{x})$, (b) the same in the $\phi$-domain, (c) a possible perturbation $\boldsymbol{z}^*$ in the $\phi$-domain by perturbing $\boldsymbol{a}$ but maintaining $\boldsymbol{e}$, and (d) the result in the image domain.

## 2.1 PROBLEM STATEMENT

Let $\phi$ be an image transformation that maps a pixel image $\boldsymbol{x} \in \mathbb{R}^n$ to a meaningful representation of the same dimension $\boldsymbol{z} = \phi(\boldsymbol{x})$. We assume that $\phi$ is bijective and (almost everywhere) differentiable. We aim to perturb some coordinates of $\boldsymbol{z}$ while leaving others unperturbed. To do so we define the split operator $\sigma$ that divides $\boldsymbol{z}$ into two vectors: $\boldsymbol{e} \in \mathbb{R}^p$ (called essential) collects the coordinates to be maintained, and $\boldsymbol{a} \in \mathbb{R}^q$ (called auxiliary) those to be perturbed. Thus, $p + q = n$. Formally,

$$\sigma(\boldsymbol{z}) = (\boldsymbol{e}, \boldsymbol{a}) \quad \text{(split)}, \quad \boldsymbol{z} = \sigma^{-1}(\boldsymbol{e}, \boldsymbol{a}) = \gamma(\boldsymbol{e}, \boldsymbol{a}) \quad \text{(combine)}. \tag{1}$$

For example, if $\phi$ is the DCT at the heart of JPEG compression, $\boldsymbol{e}$ could collect the lowest frequencies that are most important for image recovery and $\boldsymbol{a}$ the remaining higher ones.

Let $f$ be a classification model (e.g., a neural net) that correctly predicts the label $c$ of the image $\boldsymbol{x}$. After transforming $\boldsymbol{x}$ to $\phi(\boldsymbol{x}) = \boldsymbol{z}$ and applying a chosen $\sigma$ to obtain $\boldsymbol{e}$ and $\boldsymbol{a}$, we aim to perturb $\boldsymbol{a}$ to $\boldsymbol{a}^*$ such that $\boldsymbol{x}^* = \phi^{-1}(\gamma(\boldsymbol{e}, \boldsymbol{a}^*))$ gets misclassified: $f(\boldsymbol{x}^*) = c^* \neq c$ (top row in Fig. 1). The set of these perturbations yields a (not necessarily linear) subspace of dimension $q$ in the image (pixel) domain. In addition, we impose an $L^\infty$ constraint on these perturbations in the image domain.

In summary, the perturbation space we consider is given by the intersection

$$\mathbb{S}_\phi(\boldsymbol{x}) = \phi^{-1}(\gamma(\boldsymbol{e}, \mathbb{R}^n)) \cap \mathbb{B}(\boldsymbol{x}, \epsilon) \tag{2}$$

and shown in yellow in Fig. 1a with the box depicted in green. A possible perturbation $\boldsymbol{x}^*$ is shown Fig. 1d. In the transformed $\phi$-domain, $\phi(\mathbb{B}(\boldsymbol{x}, \epsilon))$ has some irregular shape (Fig. 1b), whereas the perturbations of $\boldsymbol{a}$ constitute a linear subspace.

## 2.2 ATTACK DESCRIPTION

The only free parameter in our perturbation space is $\boldsymbol{a}^*$. Thus, finding the corresponding adversarial example $\boldsymbol{x}^* = \phi^{-1}(\gamma(\boldsymbol{e}, \boldsymbol{a}^*))$ amounts to solving a constrained optimization problem of the form

$$\min_{\boldsymbol{a}^* \in \mathbb{A}} \mathcal{L}(\boldsymbol{a}^*), \tag{3}$$

where $\mathcal{L}$ is a function that promotes misclassification when minimized. Several examples have been used in the literature (Carlini & Wagner, 2017). We use the negative cross entropy $-H$:

$$\mathcal{L}(\boldsymbol{a}^*) = -H(f(\phi^{-1}(\gamma(\boldsymbol{e}, \boldsymbol{a}^*))), c). \tag{4}$$

The set $\mathbb{A} \subset \mathbb{R}^q$ in (3) represents the allowed perturbation of $\boldsymbol{a}^*$ depicted in Fig. 1b. Formally,

$$\mathbb{A} = \{\boldsymbol{a}^* \in \mathbb{R}^q : \gamma(\boldsymbol{e}, \boldsymbol{a}^*) \in \phi(\mathbb{B}(\boldsymbol{x}, \epsilon))\} = \{\boldsymbol{a}^* \in \mathbb{R}^q : \|\phi^{-1}(\gamma(\boldsymbol{e}, \boldsymbol{a}^*)) - \boldsymbol{x}\|_\infty \le \epsilon\}. \tag{5}$$

Solving this problem by a projected gradient descent (PGD) scheme, analogous to (Madry et al., 2018), would amount to iterating over two phases: updating $\boldsymbol{a}^*$ in the direction that minimizes $\mathcal{L}$ to promote misclassification and then projecting the updated $\boldsymbol{a}^*$ back into $\mathbb{A}$ as illustrated in Fig. 2a. Unfortunately, deriving this needed projection is practically unfeasible due to irregular shape of the perturbation shape for $q \geq 2$.[1] Thus, we need a fundamentally different approach to solve (3).

## 2.3 THE BARRIER METHOD

To remove the need for projection, we propose a method entirely different from the PGD approach from (Madry et al., 2018). It is based on the so-called barrier method from nonlinear programming (Chachuat, 2007; Nocedal & Wright, 2006). In the context of adversarial attacks, the barrier method was used before by (Finlay et al., 2019) to enforce a decision boundary constraint, which is fundamentally different from the subspace constraint that we are targeting.

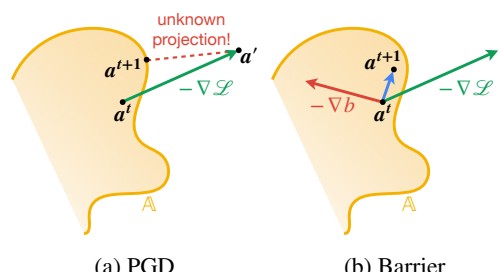

(a) PGD        (b) Barrier

Figure 2: Comparison between one update step of PGD vs. the barrier method.

To apply the barrier method, we first rewrite (3) into an inequality to obtain the standard form of nonlinear programming problems. This is straightforward using the definition of $\mathbb{A}$ in (5):

$$\min_{\boldsymbol{a}^* \in \mathbb{R}^q} \mathcal{L}(\boldsymbol{a}^*) \text{ subject to } g(\boldsymbol{a}^*) \leq 0, \text{ where } g(\boldsymbol{a}^*) = \left\| \phi^{-1}\big(\gamma(\boldsymbol{e}, \boldsymbol{a}^*)\big) - \boldsymbol{x} \right\|_\infty - \epsilon. \tag{6}$$

**Problem translation.** The barrier method translates problem (6) into the form

$$\min_{\boldsymbol{a}^*} \theta(\mu) \quad \text{s.t. } \mu \geq 0, \tag{7}$$

where $\theta(\mu) = \inf\{\mathcal{L}(\boldsymbol{a}^*) + \mu b(\boldsymbol{a}^*)) : g(\boldsymbol{a}^*) < \boldsymbol{0}\}$. The barrier function $b$ is intended to take the value zero on $\mathbb{A}$, and the value $\infty$ on its boundary. This guarantees that $\boldsymbol{a}^*$ does not leave $\mathbb{A}$, and consequently the solution $\boldsymbol{x}^*$ does not leave $\mathbb{S}_\phi(\boldsymbol{x})$ provided that the minimization problem starts at an interior point. However, this discontinuity poses difficulties for gradient-based solvers. Therefore, a more realistic construction of $b$ would be non-negative and continuous inside $\mathbb{A}$ and approach infinity as the boundary of $\mathbb{A}$ is approached. We adopt this choice: $b(\boldsymbol{a}^*) = -\frac{1}{g(\boldsymbol{a}^*)}$.

As a result, if we minimize the function $\mathcal{L}(\boldsymbol{a}^*) + \mu b(\boldsymbol{a}^*)$ starting from a point in the interior of $\mathbb{A}$, the term $b(\boldsymbol{a}^*)$ approaches infinity as $\boldsymbol{a}^*$ moves near the boundary preventing the violation of the constraint $g(\boldsymbol{a}^*) \leq \boldsymbol{0}$.

**The concrete algorithm.** Usually, the minimization in (7) is performed by a second-order Newton or quasi-Newton solver (Chachuat, 2007). However, we opt for a fast first-order update rule, which we found more practical in our setting (after setting $\boldsymbol{a}^0$ to $\boldsymbol{a}$):

$$\boldsymbol{a}^{t+1} = \boldsymbol{a}^t - \eta \cdot \text{sign}\big(\nabla_{\boldsymbol{a}} \mathcal{L}(\boldsymbol{a}^t) + \mu \nabla_{\boldsymbol{a}} b(\boldsymbol{a}^t)\big). \tag{8}$$

The idea of this update is that the gradient of the barrier function $\nabla_{\boldsymbol{a}} b(\boldsymbol{a}^t)$ pushes back when $\boldsymbol{a}^t$ approaches the boundary of $\mathbb{A}$ from the interior (see Fig. 2b for an illustration). Since this gradient has very small values on points that are far from the boundary (as $b$ is flat around the center of $\mathbb{A}$), the step size $\eta$ should therefore be small enough to allow $\boldsymbol{a}^{t+1}$ to progress slowly toward the boundary where $\nabla_{\boldsymbol{a}} b(\boldsymbol{a}^t)$ shows its effect, instead of causing a large leap that might drive $\boldsymbol{a}^{t+1}$ out of $\mathbb{A}$ as in PGD.

After $T$ iterations, we report the modified image found in this subspace $\boldsymbol{x}^* = \phi^{-1}(\gamma(\boldsymbol{e}, \boldsymbol{a}^T))$. Just as in the original PGD attack, there is no optimality guarantee of this solution for an arbitrary classifier $f$.

**Dealing with the discontinuity of the $L^\infty$ norm.** Computing the gradient of the barrier function $b$ in the iterative update (8) using the chain rule involves computing the gradient of $g$, i.e., the gradient

---

[1]Fig. 1b is misleading here since $\mathbb{A}$ has only one dimension: $q = 1$.

of the $L^\infty$ norm $\nabla||.||_\infty$. The latter is highly sparse as only one dimension is +1 or -1 (the one with the maximum absolute value) and all the other dimensions are 0. As a result, using it during optimization causes oscillation issues leaning to a poor convergence (see Sec.VI.C of (Carlini & Wagner, 2017) for a numerical example).

In our work, we eliminate this issue by replacing $g$ with another function $\tilde{g} : \mathbb{R}^q \to \mathbb{R}^{2n}$ that equivalently characterizes the set $\mathbb{A} = \{a^* \in \mathbb{R}^q : \tilde{g}(a^*) \leq 0\}$ and is defined as follows:

$$\tilde{g}(a^*)_k = \begin{cases} \phi^{-1}(\gamma(e, a^*))_k - x_k - \epsilon, & \text{for } k \leq n, \\ -\phi^{-1}(\gamma(e, a^*))_{k-n} + x_{k-n} - \epsilon, & \text{otherwise.} \end{cases}$$

Hence, the barrier function $b$ is replaced by $\tilde{b}(a^*) = -\sum_{k=1}^{2n} \frac{1}{\tilde{g}(a^*)_k}$ and then used in (8).

In the implementation, we also consider the natural range of pixels $[0, 1]^n$. That is by enforcing the inequalities $x_k \geq 0$ and $x_k \leq 1$ for all $k = 1, .., n$ through the same procedure detailed above.

# 3 INSTANTIATION FOR DIFFERENT IMAGE REPRESENTATIONS

Our attack can be used with any image transformation $\phi$ that satisfies the conditions stated in Section 2.1 and any choice of split operator $\sigma$. In this paper we consider three instantiations of $\phi$: the two classical linear DCT and DWT from the JPEG and JPEG2000 standards (Wallace, 1992; Adams, 2001), and a learned transform based on the flow-based model Glow.

**DCT.** As in JPEG, we apply the DCT on $8 \times 8$ blocks and first convert from RGB to the YCbCr color space[2]. We determine the auxiliary $a$ in the DCT domain by inspecting the JPEG quantization tables; namely $a$ collects the frequencies that are most severally reduced in the JPEG compression step (entries with large values in the quantization table, refer to Appendix A). Those are 12 out of 64 luminance frequencies and 51 out of 64 chrominance frequencies for each block. The others are assigned to the essential $e$. As a result, $p \approx 0.4n$.

**DWT.** As for the DCT, and in JPEG2000, Similarly, we first perform a color conversion before applying the two-dimensional wavelet to the entire image. As DWT we use the Cohen-Daubechies-Feauveau (CDF) 9/7 lowpass and highpass filters [3]. The result is a downscaled version of the image that we consider the essential $e$ of dimension $p = n/4$, plus horizontal, vertical, and diagonal details that we assign to the auxiliary $a$.

**Glow.** Many deep learning techniques provide meaningful representation of images such as variational auto-encoders (VAEs) (Kingma & Welling, 2013). As a third instantiation for our attack, we chose the flow-based model Glow (Kingma & Dhariwal, 2018) because it is bijective with an exact formula for the inverse, unlike other flow-based models for which one can compute the inverse only iteratively such as iResNets (Behrmann et al., 2018).

Glow is a normalizing flow (Papamakarios et al., 2021), a sequence of invertible mappings that transform images $x \in \mathbb{R}^n$ drawn from a complex intractable probability distribution, that is accessed through sampling, to latent vectors with the same dimension belonging to a Gaussian distribution $z \in \mathbb{R}^n$.

Even when Glow is trained only on images without labels, the latent space has been shown to be useful for down-stream tasks (Kingma & Dhariwal, 2018; Peychev et al., 2022). Further, we are particularly interested in the class-conditional variant of Glow (Kingma & Dhariwal, 2018), where a classification loss is introduced to effectively predict the label of the input image using only one quarter of components of the latent vector $z$. This can be viewed as a way to force these component to contain the most essential features needed to identify objects within images. More details about how we trained this model are provided in Section 4. The essential $e$ collects the aforementioned quarter of $z$ with $p = n/4$, while the rest is the auxiliary $a$.

# 4 EXPERIMENTAL EVALUATION

---

[2]Y is the luminance component and Cb and Cr are the chrominance components of the blue and red difference

[3]as defined in `https://ch.mathworks.com/help/wavelet/ref/dwtfilterbank.html`

| | Proposed attacks on the subspaces $\mathbb{S}_\phi$ | | | Baseline attacks on the full box $\mathbb{B}$ | | | | |
|---|---|---|---|---|---|---|---|---|
| | *barrier-glow* | *barrier-dwt* | *barrier-dct* | *barrier* | *pgd* | *apgd-ce* | *apgd-dlr* | *square* |
| $\epsilon = 0.025$ | | | | | | | | |
| avg. $L^\infty$ | 0.003089 | 0.0137 | 0.01138 | 0.02282 | 0.025 | 0.025 | 0.025 | 0.02447 |
| avg. $L^2$ | 0.02929 | 0.1752 | 0.117 | 0.7427 | 0.9015 | 1.134 | 1.012 | 1.346 |
| avg. LPIPS | 7.371e-06 | 1.62e-05 | 2.903e-06 | 0.0007075 | 0.0008824 | 0.001483 | 0.0009306 | 0.00866 |
| success rate (%) | 1.82 | 44.64 | 26.38 | 89.06 | 100 | 100 | 100 | 97.88 |
| $\epsilon = 0.05$ | | | | | | | | |
| avg. $L^\infty$ | 0.01043 | 0.03459 | 0.03118 | 0.04995 | 0.05 | 0.05 | 0.05 | 0.04999 |
| avg. $L^2$ | 0.09889 | 0.4258 | 0.327 | 1.465 | 1.504 | 2.07 | 1.994 | 2.738 |
| avg. LPIPS | 5.066e-05 | 7.237e-05 | 1.546e-05 | 0.002828 | 0.002967 | 0.006253 | 0.004671 | 0.03012 |
| success rate (%) | 4.09 | 77.35 | 50.55 | 99.86 | 100 | 100 | 100 | 99.98 |
| $\epsilon = 0.1$ | | | | | | | | |
| avg. $L^\infty$ | 0.0345 | 0.06623 | 0.06067 | 0.1 | 0.1 | 0.1 | 0.1 | 0.1 |
| avg. $L^2$ | 0.3329 | 0.759 | 0.6359 | 2.554 | 2.579 | 3.838 | 3.898 | 5.425 |
| avg. LPIPS | 0.0003574 | 0.000223 | 5.293e-05 | 0.01036 | 0.01055 | 0.02591 | 0.02164 | 0.07852 |
| success rate (%) | 9.41 | 97.5 | 81.54 | 100 | 100 | 100 | 100 | 100 |
| $\epsilon = 0.15$ | | | | | | | | |
| avg. $L^\infty$ | 0.06477 | 0.09185 | 0.08248 | 0.15 | 0.15 | 0.15 | 0.15 | 0.15 |
| avg. $L^2$ | 0.6381 | 1.01 | 0.8672 | 3.558 | 3.581 | 5.547 | 5.726 | 8.045 |
| avg. LPIPS | 0.001051 | 0.0004038 | 9.732e-05 | 0.02137 | 0.02159 | 0.05413 | 0.04825 | 0.1303 |
| success rate (%) | 15.53 | 99.82 | 93.95 | 100 | 100 | 100 | 100 | 100 |
| $\epsilon = 0.2$ | | | | | | | | |
| avg. $L^\infty$ | 0.09581 | 0.1141 | 0.1008 | 0.2 | 0.2 | 0.2 | 0.2 | 0.2 |
| avg. $L^2$ | 0.9616 | 1.224 | 1.066 | 4.53 | 4.552 | 7.171 | 7.616 | 10.58 |
| avg. LPIPS | 0.002097 | 0.0006111 | 0.0001472 | 0.03471 | 0.03506 | 0.0849 | 0.08265 | 0.1774 |
| success rate (%) | 21.95 | 99.98 | 97.94 | 100 | 100 | 100 | 100 | 100 |

Table 1: Evaluation of our three proposed attacks against five baseline attacks for various $L^\infty$ box radii $\epsilon$ on the correctly classified images of CIFAR10 testset (9546 images) using $T = 30$ iterations. We show the average $L^\infty$, $L^2$, and LPIPS distance of the obtained adversarial examples compared to the original (lower is better). Further, we show the success rate of the attacks. An analysis of the similarity-success rate trade-off is provided in Appendix. B

In this section, we first examine our adversarial attack on naturally trained classification models, namely a DenseNet121 (Huang et al., 2016) for CIFAR-10 and a vision transformer (ViT-B-16) (Dosovitskiy et al., 2020) for ImageNet. Since our attacks operate within predefined $L^\infty$ boxes, we compare them against state-of-the-art $L^\infty$-based attacks. Then we leverage our attacks for adversarial training and evaluate the robustness of the obtained networks against common image corruptions. All of our code and scripts to reproduce the experiments will be made available under a GPLv2 license.

The three instantiations of our method from Section 3 are called *barrier-dct*, *barrier-dwt*, and *barrier-glow* with associated perturbation spaces (see (2)) $\mathbb{S}_{dct}$, $\mathbb{S}_{dwt}$, and $\mathbb{S}_{glow}$, respectively. We also implemented an attack *barrier* in the pixel domain with standard box constraint based on the barrier method as a sanity check for comparison to PGD. Following (Kingma & Dhariwal, 2018), the class-conditional Glow architecture is composed of 3 flow levels; the depth of each level is 32, trained on CIFAR-10 for 1600 epochs with a batch size of 512.

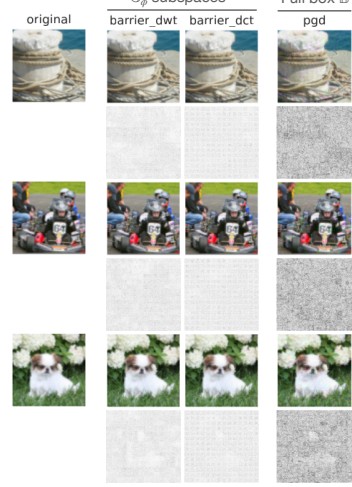

Figure 3: A sample from ImagetNet under the same settings as Fig. 4 (the Glow instantiation is omitted due to the high training cost on 256x256 images).

## 4.1 COMPARISON OF ATTACKS

We compare our attacks against four baselines: standard PGD, the two variants of the automatic projected gradient descent attack (APGD) (Croce & Hein, 2020), and the square attack (Andriushchenko et al., 2020). We ran all the mentioned attacks on the correctly classified images of CIFAR-10 testset (9546 images) and report averages in Table 1 for various choice of box bounds $\epsilon$, showing $L^\infty$ distance, $L^2$ distance, attack success rates and distance using the LPIPS similarity metric that relies on deep features learned in supervised/self-supervised/unsupervised regimes proven effective in capturing similarity between images (Zhang et al., 2018).

Further, we randomly selected images to show the found adversarial examples and the associated difference to the original in Figs. 4 (CIFAR-10) and 3 (ImageNet). Finally, for $\epsilon = 0.1$, Fig. 5 shows the interplay between LPIPS distance, $L^\infty$ distance, and attack success rate.

First, Fig. 4 visually shows that the adversarial images found by our attacks are significantly less visually impacted compared to those reported by the pixel-based attacks, even when targeting relatively large $L^\infty$ box radii. Table 1 confirms this higher visual similarity by a consistently lower values in both LPIPS similarity distance and $L^2$ distance across all experiments. We also observe that attacks based on Glow produce results that are adapted to the depicted scenery. This is not the case for DCT and DWT which modulate details. Fig. 3 shows the same behavior on image net where we consider in the instantiation of our method for classical transforms *barrier-dct* and *barrier-dwt* without the learning-based instantiation *barrier-glow* due the high cost of training this generative model on 256 by 256 images (even experiments of the paper proposing Glow (Kingma & Dhariwal, 2018) down-scaled ImageNet images to 32x32 or 64x64). We notice that *barrier-dct* introduces block boundary artifacts. In

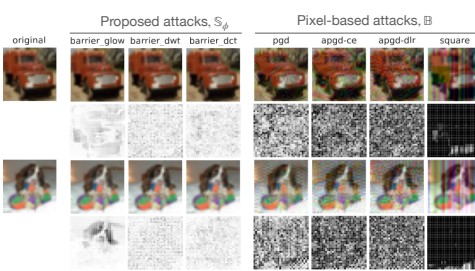

Figure 4: Two randomly selected images from CIFAR-10 and adversarial examples found by our proposed attacks compared to prior attacks, all run with $L^\infty$ box radius value of $\epsilon = 0.1$. Each row of adversarial images is followed by a row of heatmaps representing the pixelwise difference w.r.t the corresponding clean image in the first column: white = 0 and black = $\epsilon$.

the $L^\infty$ metric we observe that our proposed attacks use the freedom provide by $\epsilon$ but, unlike all benchmarks, typically do not find adversarial examples at the boundary of the box. This is also explained by the lower dimension of the perturbation subspace that we consider, and is not an intrinsic consequence of using the barrier method, since, when applied in the image domain on the entire box (our sanity check, first column of baseline attacks in Table 1), it operates similar to the PGD attack.

## 4.2 ADVERSARIAL TRAINING FOR PRACTICAL ROBUSTNESS

The high similarity and semantic nature (in the sense of the transform being used) of the adversarial examples produced by our attacks motivates their use as proxies to achieve robustness against another class of image perturbations that preserves visual similarity: common image corruptions. To do so, we use our attacks for adversarial training (AT), a technique in which the neural network is trained on adversarial examples aiming to increase robustness against this adversary.

Specifically, we adopt the AT technique TRADES (Zhang et al., 2019), which is a heuristic algorithm based on multi-class calibrated loss theory that balances the trade-off between robustness and accuracy. We train for robustness under four types of constraints: $L^\infty$ box $\mathbb{B}$, and DCT subspace $\mathbb{S}_{\text{dct}}$, DWT subspace $\mathbb{S}_{\text{dwt}}$ and Glow subspace $\mathbb{S}_{\text{glow}}$ against their corresponding adversaries: the standard *pgd*, *barrier-dct*, *barrier-dwt*, and *barrier-glow*, respectively. All ATs are granted the same number of iterations $T = 10$ to fetch adversarial examples for each training epoch using a common choice of $\epsilon = 0.05$. The naturally trained vanilla model is the same as Sec. 4.1. All models are trained without any additional data. All models are DensNet121 (Huang et al., 2016) which is a multigrade architecture found to resist noise corruptions more effectively than ResNets (Hendrycks & Dietterich, 2019).

We consider the CIFAR-10-C dataset (Hendrycks & Dietterich, 2019), a benchmark constructed by applying common image corruptions to the CIFAR-10 test set. These corruptions are only used for evaluation and not to augment the data during training. The results for all categories in CIFAR-10-C are reported in Table 2. The first data column show the accuracy on the clean, unperturbed images. The latter columns show the accuracy on corrupted images, considering 18 different types of corruptions: different forms of blurring, digital, noise corruptions, and weather related ones.

First, networks trained under our subspace constraints achieve higher accuracy on corrupted images compared to the model trained under the box constraint across all types of corruptions. They do so while suffering only a minor reduction in natural accuracy compared to the vanilla network. The performance of AT with our DWT subspace constraint $\mathbb{S}_{dwt}$ stand out in particular. On average, it is

Table 2: Accuracies on different image corruption categories of networks adversarially trained under different constraints.

| | Natural | Blur | | | | Digital | | | | |
| --- | --- | --- | --- | --- | --- | --- | --- | --- | --- | --- |
| | | Gauss | Motion | Defocus | Zoom | Contrast | Elastic | JPEG | Pixel | Saturate |
| Vanilla (no AT) | **95.46** | 72.36 | 81.22 | 83.81 | 77.74 | **83.01** | 85.04 | 81.07 | 74.83 | **92.46** |
| Full $L^\infty$ box $\mathbb{B}$ | 81.23 | 75.36 | 73.77 | 77.12 | 76.30 | 45.47 | 75.54 | 79.30 | 79.23 | 77.75 |
| Proposed subspace $\mathbb{S}_{\text{dct}}$ | 94.84 | 76.36 | 79.06 | 85.01 | 80.81 | 78.45 | 86.06 | 88.02 | 77.13 | 92.23 |
| Proposed subspace $\mathbb{S}_{\text{dwt}}$ | 93.83 | **87.95** | **85.55** | **89.91** | **88.79** | 67.28 | **88.42** | **89.44** | **92.09** | 88.93 |
| Proposed subspace $\mathbb{S}_{\text{glow}}$ | 95.41 | 77.56 | 84.27 | 85.93 | 81.75 | 82.81 | 86.73 | 82.36 | 75.31 | 92.23 |

| | Natural | Noise | | | | Weather | | | | |
| --- | --- | --- | --- | --- | --- | --- | --- | --- | --- | --- |
| | | Gauss | Impulse | Speckle | Shot | Snow | Fog | Frost | Bright | Spatter |
| Vanilla (no AT) | **95.46** | 48.50 | 60.21 | 65.09 | 61.09 | 84.33 | 89.15 | 81.55 | **94.00** | 87.99 |
| Full $L^\infty$ box $\mathbb{B}$ | 81.23 | 75.61 | 73.78 | 76.67 | 76.68 | 75.15 | 59.69 | 71.02 | 77.92 | 77.24 |
| Proposed subspace $\mathbb{S}_{\text{dct}}$ | 94.84 | 68.83 | 68.71 | 76.56 | 75.83 | 85.37 | 86.84 | 83.60 | 93.62 | 87.86 |
| Proposed subspace $\mathbb{S}_{\text{dwt}}$ | 93.83 | **81.83** | **73.98** | **84.93** | **85.24** | **88.89** | 80.34 | **88.79** | 92.05 | **89.78** |
| Proposed subspace $\mathbb{S}_{\text{glow}}$ | 95.41 | 43.77 | 57.32 | 60.33 | 56.15 | 85.50 | **90.81** | 82.44 | 93.99 | 87.28 |

7.31% more accurate than the vanilla model and 12.27% more accurate than the model trained under the standard box constraint $\mathbb{B}$. The sacrifice in natural accuracy is only about 1.5%. We note that these results are in line with recent advances in machine learning interpretability where the wavelet domain also provides better performance than the pixel-based methods (Kolek et al., 2022).

Finally, we note that AT with $\mathbb{S}_{glow}$ does not perform well on corruption, possibly since the features learned by the flow-based model are semantically at a higher level, and thus not compatible with the considered corruptions that are closer related to the DCT and DWT frequency representations. However, AT with $\mathbb{S}_{glow}$ practically maintains the natural accuracy.

**Limitations and discussion.** For the experiment in Table 2 there are techniques that achieve better corruption robustness. These are not based on adversarial attacks but on other data augmentation techniques (Geirhos et al., 2018; Erichson et al., 2022; Zhang et al., 2017; Hendrycks et al., 2019; Park et al., 2022; Yin et al., 2022; Liang et al., 2023). They are specifically targeting this benchmark (whereas our approach is oblivious to it) and usually train substantially larger networks (e.g., WideResNet-28-4 used by NoisyMix (Erichson et al., 2022)) and require pre-training on larger datasets.

Our goal was to expand the tool set of adversarial attacks and to also make progress on the link between adversarial robustness and corruption robustness as a followup to the findings of (Hendrycks & Dietterich, 2019; Ford et al., 2019; Xie et al., 2020; Kang et al., 2019; Kireev et al., 2022). The precise specification of our perturbation space makes porting of state-of-the-art certification techniques (either approximation-based (Singh et al., 2019; Müller et al., 2022) or probabilistic (Cohen et al., 2019; Carlini et al., 2022)) to operate under the proposed constraints possible. The generality of our approach in the choice of transform $\phi$ and associated subspace to be perturbed invites further exploration.

## 5 RELATED WORK

We cited a number of related work in the introduction and throughout the paper. Here we focus on prior uses of transformed image representations. In particular, discrete linear transforms have been used in machine learning for different purposes. For example, (Gueguen et al., 2018; dos Santos & Almeida, 2021) proposed DCT-based architectures operating directly on the JPEG format to avoid decompression before inference. Furthermore, (Kolek et al., 2022) have extended the rate-distortion framework (MacDonald et al., 2019) to the wavelet domain to build a state-of-the-art explanation method for DNN. The remainder of this section is focused on previous works related to robustness.

**Discrete transforms for robustness.** Most prior work using discrete transforms aimed at defending against pixel-based adversarial attacks or improving the generalization of neural networks towards common image corruptions. The work of (Dziugaite et al., 2016; Das et al., 2017; Guo et al., 2018) aims to filter out noise from the adversarial examples by adjusting various quality factor values during JPEG compression/decompression, which amounts to reducing the magnitude of the DCT coefficients. Closely related, (Bafna et al., 2018) sought $L^0$ robustness through projecting the

largest DCT coefficients. These defenses have been shown to be breakable through adaptive attacks (Shin & Song, 2017; Tramèr et al., 2020), specifically, by approximating the non-differentiable rounding operator of the JPEG compression and running a gradient-based attack. Other fast and iterative rounding schemes have been proposed in (Shi et al., 2021b). (Yin et al., 2019; Guo et al., 2019) considers $L^2$ perturbations that preserve norms due to orthogonality of the used transforms, discrete Fourier transform (DFT) and DCT respectively. The work in (Duan et al., 2021) generates adversarial examples by removing information in the DCT domain. The $L^\infty$ box used is on the JPEG quantization matrix instead of the input image. Since the DCT coefficients of the clean image are element-wise divided by this matrix before rounding, larger box radii allow their technique to eliminate more frequencies from the image. In the same direction, (Hossain et al., 2019) preceded the neural network by a DCT based layer that randomly crops some DCT coefficients during training. This can be interpreted as an extension of the dropout technique aiming at its regularization effects. (Yahya et al., 2020) propose a gradient-free method that obtains adversarial examples by mixing the frequencies of a clean image with the frequencies of another auxiliary image that they call watermark. In addition to DFT and DCT, they make use of two wavelets: Haar and Daubechies 3. (Sharma et al., 2019) applies masks to selectively perturb low and high frequencies. Much like (Deng & Karam, 2020; Shi et al., 2021a), all these works do not provide any guarantee on the $L^\infty$ bounds in the pixel space, which is the primary contribution in our work. We can also target low dimensional spaces which (Long et al., 2022) cannot. Yuan et al. (2022) proposes a DCT-based attack and uses the fact that DCT is linear and orthogonal, where our method only needs invertible and differentiable (e.g. Glow) since we do not need projections due to the barrier method. (Luo et al., 2022; Wang et al., 2021; Laidlaw et al., 2021; Kireev et al., 2022) explicitly uses a similarity distance in the optimization problem formulation in the pursuit of semantically similar adversarial examples.

In contrast to all prior work we show how to perform attacks on a subspace of an image representation, linear or not, that also enforces the $L^\infty$ box in the image domain.

**Learning-based methods for robustness.** Flow-based generative models themselves are prone to adversarial attacks that manipulate their likelihood scores (Pope et al., 2020). That is a different focus from our work as we are used a flow-based generative model (in one of our 3 instances) to define a meaningful subspace rather than a likelihood estimation. (Huang & Zhang, 2020; Baluja & Fischer, 2018) trained NNs to produced perturbations under the $L^\infty$ and $L^2$ constraints that can operate in the black-box settings. Adversarial generative models (GANs) also have been trained to generate adversarial examples in semi-whitebox and black-box settings. (Dolatabadi et al., 2020) used a pre-trained flow-based model (RealNVP (Dinh et al., 2017)) to craft adversarial examples in black-box settings under the $L^\infty$ constraint. They used an additive noise in the latent space where they faced a similar projection problem as this work. They solve it by going back and forth to the pixel space to project using the PGD formula. In contrast, we removed the projection by using the barrier method in a way that is fast enough to be incorporated in adversarial training. More importantly, we operate in a subspace within the box instead of the full box. We show that doing so is effective in AT to produce networks robust to common image corruptions. Some works altered the semantic features of images through conditional generative models (Joshi et al., 2019) or conditional image editing(Qiu et al., 2020), but those are not bounded to a norm.

## 6 CONCLUSION

We have expanded the toolbox of adversarial attacks, and associated adversarial training, with a general, and powerful novel method. The novelty is twofold. First, in the ability to attack semantically (in a sense associated with the chosen transform) in a suitable transformed image representation space while preserving proximity in the pixel space. Second, in using the barrier method needed to enable such an attack when projections are not available. Thus our approach fuses two prior lines of research that attack in either of these spaces. We emphasize that the transform used does not need to be linear, only invertible, as we show by also considering a learned transform. The benefit of our approach is best visible when used for adversarial training: in particular with the DWT a major improvement in accuracy on a broad range of corruptions with only a small drop in natural accuracy.

The generality of our work in both chosen transform and chosen subspace should invite further exploration. In particular, by leveraging decades of research on image representations for better defining constraints under which adversarial robustness is studied.

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

## A SELECTING DCT COEFFICIENTS FOR THE SPLIT OPERATOR $\sigma$

These are the quantization tables used in JPEG for the luminance $\boldsymbol{Q}_Y$ and the two chrominance channels $\boldsymbol{Q}_C$. We select the DCT coefficients corresponding to entries below 99 (in bold) to the essential vector $\boldsymbol{e}$ while the rest is the auxiliary vector $\boldsymbol{a}$.

$$
\boldsymbol{Q}_Y = \begin{bmatrix}
16 & 11 & 10 & 16 & 24 & 40 & 51 & 61 \\
12 & 12 & 14 & 19 & 26 & 58 & 60 & 55 \\
14 & 13 & 16 & 24 & 40 & 57 & 69 & 56 \\
14 & 17 & 22 & 29 & 51 & 87 & 80 & 62 \\
18 & 22 & 37 & 56 & 68 & 109 & 103 & 77 \\
24 & 35 & 55 & 64 & 81 & 104 & 113 & 92 \\
49 & 64 & 78 & 87 & 103 & 121 & 120 & 101 \\
72 & 92 & 95 & 98 & 112 & 100 & 103 & 99
\end{bmatrix}
\tag{9}
$$

$$
\boldsymbol{Q}_C = \begin{bmatrix}
17 & 18 & 24 & 47 & 99 & 99 & 99 & 99 \\
18 & 21 & 26 & 66 & 99 & 99 & 99 & 99 \\
24 & 26 & 56 & 99 & 99 & 99 & 99 & 99 \\
47 & 66 & 99 & 99 & 99 & 99 & 99 & 99 \\
99 & 99 & 99 & 99 & 99 & 99 & 99 & 99 \\
99 & 99 & 99 & 99 & 99 & 99 & 99 & 99 \\
99 & 99 & 99 & 99 & 99 & 99 & 99 & 99 \\
99 & 99 & 99 & 99 & 99 & 99 & 99 & 99
\end{bmatrix}
\tag{10}
$$

## B THE SIMILARITY-SUCCESS RATE TRADE-OFF

Adversarial examples exhibit high similarity. The trade-off for this higher similarity is a lower success rate for very small $\epsilon$, whereas the prior benchmarks almost always succeed as visualized in Fig. 5. This is due to the fact that, despite being bounded by the same $L^\infty$, our methods operate only on a $q$-dimensional subspace of the $n$-dimensional box, where $q \approx 0.6n$ for *barrier-dct*, and $q = 0.75n$ for *barrier-dwt* and *barrier-glow*.

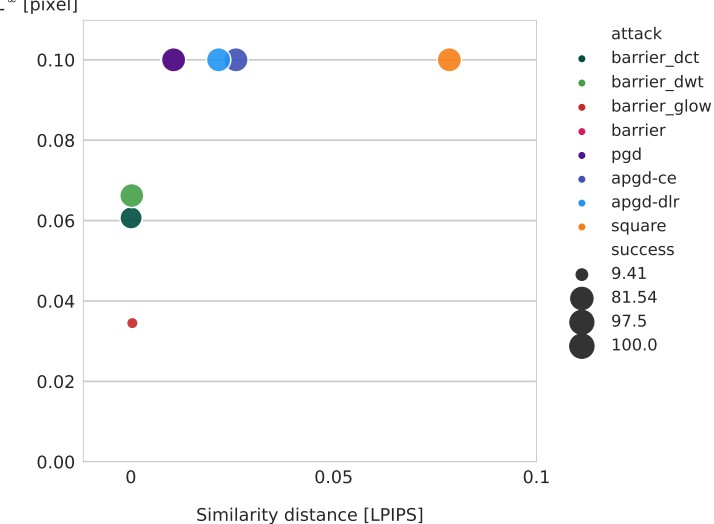

Figure 5: Interplay of LPIPS distance, $L^\infty$ distance, and success rate (encoded by marker size) for $\epsilon = 0.1$ based on the numbers in Table 1.

