# OpenReview forum: "Leveraging image representations for bounded adversarial attacks and robustness"
_ICLR.cc/2024/Conference — Submitted to ICLR 2024_

### Official Review · Reviewer_F4jF · 2023-11-08

**Soundness:** 3 good
**Presentation:** 3 good
**Contribution:** 3 good
**Rating:** 5
**Confidence:** 4

**Summary:**

This paper introduces a novel adversarial attack which relies on perturbing images in representation space such that, when this representation is inverted, we get an adversarial example in the pixel space. To this end, the authors create a framework where given an invertible transformation (DCT, DWT and GLOW are used in the paper), we can put an image through this transform and then separate out the most significant components of its representation. These important components are kept the same while the remaining components (presumed to be less important) are perturbed slightly. The space of these perturbations is such that when the perturbed representation is inverted, we get an adversarial example in pixel space (w.r.t the $\infty$ norm).

To actually produce the adversarial examples, the authors follow a method similar to PGD except that they replace the ‘projection’ step in PGD by placing a constraint in their optimization problem. This is just the standard barrier function method from constrained optimization which makes sure that the iterates of the optimization remain within the set of possible outcomes (within the set of allowed perturbations in this case).

This, the authors show, produces adversarial examples which are closer to the original image in terms of L2, LPIPS etc distance. Next, the authors use their attack for adversarial training which helps the model robustify against image corruptions (such as blue, weather etc) with small losses in clean accuracy. Since the adversarial attacks were not tuned w.r.t these image corruptions, the authors show that their attack introduces robustness which transfers over to other tasks (image corruptions in this case).

**Strengths:**

1. Significance of the problem: robustifying models against unseen threat models is an extremely important problem facing our community. The authors attempt to make a preliminary advance in pursuit of that goal.
2. The method introduced in the paper is novel to my knowledge and well motivated.
3. The paper is well written and easy to understand. Novel concepts introduced are explained well and their relation to past work is explained clearly.
4. The claims made by the authors in the introduction are clearly backed up via experiments.

**Weaknesses:**

My main concern about this paper revolves around the significance of the results.

I would break down the results into 3 parts:
1. First, the authors state that their attack produces adversarial examples which are closer to the original image than what we would get via PGD. This is measured via a number of distance metrics in pixel space (L2, LPIPS etc)
2. Second, they then show that when their attack is used for adversarial training, we get only a small drop in clean accuracy (only a few % points). This to me seems expected, because, the images produced are very similar to the original images so the model makes fewer mistakes (Point 1 above).
3. Next, the authors show that adversarial training using their attack introduces robustness to threat models which were not trained against originally. This is a very significant problem and, from what I’ve seen, usually solved via some form of ensembling. However, the robustness introduced against image corruptions via this method falls short of what dedicated methods attain (this is a limitation pointed out by the authors to be fair and not to be held against them since they use a threat model agnostic method).

For Point 3 above, while the authors have shown that their method transfers better to image corruptions than vanilla PGD, there needs to be some control for the amount of image corruption added. As stated above, adversarial examples produced by the authors are closer to the original image. Hence, it might be possible that their method only performs better because the amount of corruption added to the image is too low and if more noise was added, vanilla PGD might become better.
Additionally, I would like to see the authors compare attacks with each other. If the claim is about the transferability of robustness, why not train the model with adversarial examples produced using their own attack and test them on adversarial examples produced via PGD? (And vice versa) Regardless of the outcome of this experiment, it would tell us something about the limits of transferability being claimed.

I do think this paper contributes something meaningful methodologically (perturbing in representation space), however, it is unclear to me where or what this method is actually good at doing. This is the main reason behind my score below.

Minor:
1. There are several choices made throughout the paper which seem essential to making this technique work. It would be nice to have ablations over these choices to make the paper more rigorous experimentally. Some examples: choice of barrier function, size of ‘e’ in class conditional representations for GLOW etc.

Typos:
1. Page 1, Section “Introduction”, Subsection “Adversarial Robustness” - “attacks either by LEANING and embedding space using neural networks (Huang….)” should be “attacks either by LEARNING and embedding space using neural networks (Huang….)”
2. Page 2, Paragraph 2 - “Approaches include again suitable data augmentation” I presume should be “Approaches include using suitable data augmentation”
3. Page 5, Section 3, Subsection “DCT” - “collects the frequencies that are most SEVERALLY reduced” should be “collects the frequencies that are most SEVERELY reduced”
4. Page 5, Section 3, Subsection “DWT” - Capitalization “As for the DCT, and in JPEG2000, Similarly, we first perform” should presumably be something like “Similar to the case of DCT for JPEG2000, we first perform”. Note, this is just a suggestion and the authors might have been trying to communicate something else so feel free to use some other statement. The point of this was only to correct capitalization and grammar.
5.  I think the second argument for $\gamma$ in Eq 2 should be $R^{q}$ right? Because that is the domain of the ‘a’ values.

Formatting:
1. Title of section 4 “Experimental Evaluation” is the last line of the page. Ideally, the title should have some following text so that the document flows better.

**Questions:**

1. The barrier function was chosen to be of the form $b(a^{*}) = -\frac{1}{g(a^{*})}$. What is the reason for this choice apart from it being continuous? Were other choices considered?
2. Why is just the sign of the gradient taken in the SGD update instead of the value? Is there any reason beyond just that it works better in practice in this setting?

---

> ### Author Response · Authors · 2023-11-18
>
> > " Additionally, I would like to see the authors compare attacks with each other. If the claim is about the transferability of robustness, why not train the model with adversarial examples produced using their own attack and test them on adversarial examples produced via PGD? (And vice versa) Regardless of the outcome of this experiment, it would tell us something about the limits of transferability being claimed."
>
> Indeed this experiment is interesting. We did it and show the results below. In this table, rows correspond to classifiers trained under different regimes:
> * No adversarial training: Vanilla training,
> * AT using TRADES loss in $L^\infty$ box: TRADES in $\mathbb{B}_\infty$,
> * AT using TRADES loss in the proposed subspaces:
>     * of DCT: $\mathbb{S}_\text{dct}$,
>     * of DWT: $\mathbb{S}_\text{dwt}$,
>     * of Glow: $\mathbb{S}_\text{glow}$.
>
> Columns represent accuracies measured on:
> * clean images: **Clean acc.**,
> * average accuracy across all 19 corruption types: **Avg. acc. of 19 corruptions**,
> * accuracy in adversarial examples produced by PGD laying the $L^\infty$ box: **Adv. acc. in $\mathbb{B}_\infty$**,
> * accuracy in adversarial examples produced by our barrier attacks operating in:
>     * DCT subspace: **Adv. acc. in $\mathbb{S}_\text{dct}$**,
>     * DWT subspace: **Adv. acc. in $\mathbb{S}_\text{dwt}$**,
>     * and Glow subspace: **Adv. acc. in $\mathbb{S}_\text{glow}$**.
>
> For example, the model trained using adversarial examples produced by our barrier attack in the DCT space (third row) achieves an accuracy of 90.3\% when tested on adversarial examples produced by the barrier attack operating in the Glow space (last column).
>
>
> |                  | Clean acc. | Avg. acc. of 19 corruptions |Adv. acc. in $\mathbb{B}_\infty$ | Adv. acc. in $\mathbb{S}_\text{dct}$ | Adv. acc. in $\mathbb{S}_\text{dwt}$ | Adv. acc. in $\mathbb{S}_\text{glow}$
> | :---                |          ---: |  ---: | --: | --: | --: | --:
> | Vanilla training   | 95.46   |78.89| 0.01 | 53.1 | 32.38 | 92.09
> | TRADES on $\mathbb{B}_\infty$ | 81.23 |73.93 | 36.8 | 71.13 | 67.54 | 76.3
> | TRADES on $\mathbb{S}_\text{dct}$ (ours) | 94.84 |82.37| 0.11 | 93.1 | 64.6 | 90.3
> | TRADES on $\mathbb{S}_\text{dwt}$ (ours)| 93.83 | **86.21** | 0.39 | 88.2 | **90.1** | 89.41
> | TRADES on $\mathbb{S}_\text{glow}$ (ours)| 95.41 |79.04|  0.0 | 54.09 | 33.24 | 92.5
> |
>
> $$\def\B{\mathbb{B}_\infty}$$
>
> $$\def\Sone{\mathbb{S}_\text{dct}}$$
>
> $$\def\Stwo{\mathbb{S}_\text{dwt}}$$
>
> $$\def\Sthree{\mathbb{S}__\text{glow}}$$
>
>
> Training for robustness in $\B$  improves the robustness in this box (as expected) and further improves the robustness in tow of our proposed subspaces ($\Sone$ and $\Stwo$) but it decreases the robustness in the Glow space, on clean images and most importantly it decreases the average accuracy across the 19 corruption types. On the other hand, the models trained on images produced by our attacks (in particular the one operating the DWT subspace $\Stwo$) present low accuracy on images produced by PGD, but they maintain high accuracy on images produced by the corresponding attack (the one used for their training). Finally, we can conclude that the robustness in the proposed subspace (in particular $\Stwo$ of 90.1\%) transfers to robustness under unseen images corruptions.
>
>
> > "The barrier function was chosen to be of the form $b(a^{}) = -\frac{1}{g(a^{})}$. What is the reason for this choice apart from it being continuous? Were other choices considered?"
>
> There are two popular choices of the barrier functions in the literature, the negative inverse: $b(a) = -1/g(a)$, and the log: $b(a) = \log{g(a)}$. We have implemented both and we found that the former performs better.
>
> > "Why is just the sign of the gradient taken in the SGD update instead of the value? Is there any reason beyond just that it works better in practice in this setting?"
>
> There is no theoretical reason, only practical.

---

> > ### Comment · Reviewer_F4jF · 2023-11-23
> > **Response to rebuttal**
> >
> > I thank the authors for the additional experiments.
> >
> > I do not see how this experiment supports the authors hypothesis of transferability. In fact, it seems to show a limit on how the method transfers.
> >
> > My main concern regarding the magnitude of image corruptions also remains unaddressed.
> >
> > I will maintain my rating.

---

> > > ### Author Response · Authors · 2023-11-23
> > >
> > > > I do not see how this experiment supports the authors hypothesis of transferability. In fact, it seems to show a limit on how the method transfers.
> > >
> > > The model trained to be robust under our proposed DWT subspace (that has an adversarial accuracy of 90.1% in this subspace) is also the most robust against image corruptions (average accuracy of 19 corruptions	is 86.21). So training for robustness under the DWT space transfers to robustness against unseen 19 image corruptions. You have worded this transferability differently: "Next, the authors show that adversarial training using their attack introduces robustness to threat models which were not trained against originally. This is a very significant problem and, from what I’ve seen, usually solved via some form of ensembling."
> > >
> > >
> > > > My main concern regarding the magnitude of image corruptions also remains unaddressed.
> > >
> > > * If you mean by "magnitude of image corruptions" the severity of image corruption types taken from the CIFAR-10-C image corruption benchmark: we have used the default severity 3 (the maximum is 5).
> > > * If you mean by "magnitude of image corruptions" the difference between the clean images and the adversarial examples:
> > >
> > >     * For the attack evaluation: we have computed the $L^2$ and $L^\infty$ of the difference between clean images and adversarial examples produced by our attacks and those produces by the baseline attacks. We did so for 4 values of $\epsilon$ and reported the results in Table. 1.
> > >
> > >     * For adversarial training: we have adopted a common magnitude: an $L^\infty$ box radius $\epsilon=0.05$. This magnitude is a common choice when PGD is used for adversarial training. We show that our method (selecting subspaces inside the box based on image transformations) improves the robustness on CIFAR-10-C compared to adversarial training on the full box.

---

### Official Review · Reviewer_8uZy · 2023-11-10

**Soundness:** 3 good
**Presentation:** 3 good
**Contribution:** 3 good
**Rating:** 5
**Confidence:** 4

**Summary:**

The paper proposes to find adversarial examples by perturbing the image in the representation space and then convert it back to the original image space. Empirical experiments show that this method can produce adversarial examples with more natural adversarial examples and smaller Lipschitz distance.

**Strengths:**

1. The paper proposes to generate adversarial examples with invertible image representations. This idea is novel and interesting, which is not known in existing literature as far as I know.
2. Empirical experiments show that the proposed method can produce adversarial examples with significantly smaller Lp distances.

**Weaknesses:**

1. The attack success rate of the proposed method is much lower compared to baseline methods, which may also weakens the results with smaller perturbation radius. The author could report avg. Lp distance of the same top% of adversarial examples from the baseline method.
2. As the proposed method assume that the image representation function $\sigma$ is invertible, do the DCT and DWT implemented in the paper invertible? In my understanding they produce a compressed version of the original image and are not invertible.

**Questions:**

Please see the weaknesses.

---

> ### Author Response · Authors · 2023-11-18
>
> > "...The author could report avg. Lp distance of the same top% of adversarial examples from the baseline method."
>
> We have reported the avg. $L^p$ distances in Table 1. Would you please clarify what you mean by "same top%"?
>
> > "As the proposed method assume that the image representation function $\sigma$ is invertible, do the DCT and DWT implemented in the paper invertible? In my understanding they produce a compressed version of the original image and are not invertible."
>
> The DCT and DWTs themselves are invertible, the former is even orthogonal (L2 norm preserving). Often, they are used in a pipeline with a subsequent lossy transformation (e.g., removing high frequencies or quantization). We did not include these.

---

### Official Review · Reviewer_bTry · 2023-11-10

**Soundness:** 3 good
**Presentation:** 2 fair
**Contribution:** 2 fair
**Rating:** 3
**Confidence:** 3

**Summary:**

This paper proposes to conduct adversarial attacks in a meaningful subspace of a transformed image representation while obeying the Linf box at the same time. Since the perturbation space is more complex, PGD cannot be directly applied, and the authors proposed to use the barrier method from nonlinear programming to compute perturbations without a projection. Experiments show that the proposed method can generate adversarial images that are more similar to original images, in the Linf adversarial attack setting. Experiments also show that when the proposed method is used in adversarial training, it can improve robustness against corruptions.

**Strengths:**

* This work proposed a new threat model which considers perturbations in a meaningful subspace of a transformed image representation.
* This work designed an attack for the proposed threat model, using a barrier method.
* Experiments show that adversarial training with the proposed threat model can improve robustness against corruptions.

**Weaknesses:**

* Comparison with baseline attacks using Linf boxes is not fair. The proposed method has smaller perturbations but also lower success rates. It is possible that the perturbation size of the baseline attacks may be reduced at the cost of the success rates. The proposed method and the baseline methods need to be compared by controlling for the success rate or the perturbation size.
* The introduction says "In this paper we offer progress in the quest for corruption robustness by present- ing a powerful novel adversarial attack and associated adversarial training". Thus, it sounds like improving the robustness against corruption is the main purpose of this paper. However, the experiments do not compare with any baseline specifically for robustness against corruption.
* The proposed adversarial training has improvement on Blur and Noise, but not really on Digital and Weather. Thus, the significance of the empirical improvement is limited.
* The concept of subspace adversarial training has earlier appeared in Li et al., 2022, which is missed in this work, although the techniques are not the same.

Li, T., Wu, Y., Chen, S., Fang, K., & Huang, X. (2022). Subspace adversarial training. In Proceedings of the IEEE/CVF Conference on Computer Vision and Pattern Recognition (pp. 13409-13418).

----------------

**Updates after rebuttal**:

>We are approaching the problem of robustness from an adversarial training perspective unlike state-of-the-art corruption robust methods that are mainly based on data augmentation. We have discussed this in Section 4, the Limitations and discussion paragraph.

The problem and the goal is still the same, and thus a comparison is necessary.

>Not correct. From Table 2: under the Digital category, we have improved in Elastic from 85.04 to 88.42, in JPEG from 81.07 to 98.44, and in Pixelization from 79.23 to 92.09. In the Weather category, the improvement is in Snow from 84.33 to 88.89, in Fog from 89.15 to 90.81, and in Spatter from 87.99 to 89.78.

The improvement is unstable and inconsistent. E.g., for the Digital category, while JPEG is improved from 81.07 to 89.44, Contrast is significantly degraded from 83.01 to 67.28. Same issue for the Weather category.

Therefore, most of the weaknesses I originally mentioned persist. I'll keep my score.

**Questions:**

See the weakness points.

---

> ### Author Response · Authors · 2023-11-18
>
> > "Comparison with baseline attacks using Linf boxes is not fair. The proposed method has smaller perturbations but also lower success rates. It is possible that the perturbation size of the baseline attacks may be reduced at the cost of the success rates. The proposed method and the baseline methods need to be compared by controlling for the success rate or the perturbation size."
>
> We have considered controlling the perturbation size for a fairer comparison. However, the subspace our attacks are operating on have different dimensions than the full space (the PGD's) $q<n$. This renders the "sizes" or the volumes incomparable. The success rate is unknown until running a particular attack over a set of images which prevents us from fixing the success rate a priori for comparison. As a result, we have compared based on the $L^\infty$ radius. We are open to suggestions.
>
> > "...However, the experiments do not compare with any baseline specifically for robustness against corruption."
>
> We are approaching the problem of robustness from an adversarial training perspective unlike state-of-the-art corruption robust methods that are mainly based on data augmentation. We have discussed this in Section 4, the Limitations and discussion paragraph.
>
> > "The proposed adversarial training has improvement on Blur and Noise, but not really on Digital and Weather. Thus, the significance of the empirical improvement is limited."
>
> Not correct. From Table 2: under the Digital category, we have improved in Elastic from 85.04 to 88.42, in JPEG from 81.07 to 98.44, and in Pixelization from 79.23 to 92.09. In the Weather category, the improvement is in Snow from 84.33 to 88.89, in Fog from 89.15 to 90.81, and in Spatter from 87.99 to 89.78.
>
> > "The concept of subspace adversarial training has earlier appeared in Li et al., 2022, which is missed in this work, although the techniques are not the same."
>
> Yes, the subspaces we are proposing are induced by semantically relevant image transformations in contrast to those by Li et al., 2022 - we will add a reference.

---

> ### Comment · Reviewer_bTry · 2023-12-03
> **Thanks for the response**
>
> >We are approaching the problem of robustness from an adversarial training perspective unlike state-of-the-art corruption robust methods that are mainly based on data augmentation. We have discussed this in Section 4, the Limitations and discussion paragraph.
>
> The problem and the goal is still the same, and thus a comparison is necessary.
>
> >Not correct. From Table 2: under the Digital category, we have improved in Elastic from 85.04 to 88.42, in JPEG from 81.07 to 98.44, and in Pixelization from 79.23 to 92.09. In the Weather category, the improvement is in Snow from 84.33 to 88.89, in Fog from 89.15 to 90.81, and in Spatter from 87.99 to 89.78.
>
> The improvement is unstable and inconsistent. E.g., for the Digital category, while JPEG is improved from 81.07 to 89.44, Contrast is significantly degraded from 83.01 to 67.28. Same issue for the Weather category.
>
> Therefore, most of the weaknesses I originally mentioned persist. I'll keep my score.

---

### Official Review · Reviewer_p6XX · 2023-11-10

**Soundness:** 2 fair
**Presentation:** 3 good
**Contribution:** 2 fair
**Rating:** 5
**Confidence:** 3

**Summary:**

The authors propose a novel white box adversarial attack method by creating perturbations in a transformed image representation space while constraining the maximum pixel perturbation. To this end, the image in transformed space is split into two vectors and one of them is perturbed. This problem is then converted to a linear programming problem and the barrier method is used to find the optimal value (the adversarial example). Three different image representations - DCT, DWT and glow (2 linear and 1 learner based) are used to demonstrate the effectiveness of attack on CIFAR dataset. The attack method is then used in adversarial training (TRADES used here) to show its effectiveness in training robust models. Experiments are presented to test the robustness against various kinds of corruptions.

**Strengths:**

The idea presented in the paper is very interesting and novel. Converting the adversarial attack problem into a linear programming problem and using adequate tools to solve it is a good catch. The authors have explained well the problems encountered during making such a conversion and strategies used to overcome it. E.g. using barrier method to remove need for projection, using first-order update rule instead of the usual minimization, dealing with discontinuity of $L^{\infty}$ norm. Overall the problem is well formulated and explained.

The flow of the paper is clear and easy to understand. In terms of significance, this reformulation of adversarial attack is interesting and might be for the useful for the community to look at.

**Weaknesses:**

The idea presented is novel but the experiments do not back it up well.

1. The method finds examples that are less perturbed than other methods compared but the attack success rate is not upto par with other methods. In fact it is very low in some of the image representations used. The presented the trade-off between distance from original image and success rate does not show it to be very effective attack method. You can't really visually tell the difference between the adversarial images generated from this attack and PGD.
2. The attack methods are only compared on CIFAR - Imagenet would be a good addition as datapoint to the experiments.
3. No mention of the cost of attack, number of steps to reach the adv example and time taken by barrier method optimization.
4. For adversarial training, comparison has only been made with PGD. I believe other methods should be used in the comparison.
5. Again ImageNet not used for adversarial training, would be a good experiment to add.

Overall the experiments do not showcase the effectiveness of method as adversarial attack or for training robust models compared to other state of the art methods.

Typo:

Eqn 2 : $R^n$ -> $R^q$ ?

**Questions:**

See weaknesses

---

> ### Author Response · Authors · 2023-11-18
>
> > "The method finds examples that are less perturbed than other methods compared but the attack success rate is not upto par with other methods. In fact it is very low in some of the image representations used. The presented the trade-off between distance from original image and success rate does not show it to be very effective attack method."
>
> True, there is a trade-off since our attacks operate in low dimensional spaces (we have more constraints to satisfy in addition to the $L^\infty$ box). We have discussed this trade-off in **Appendix B**.
>
> > "You can't really visually tell the difference between the adversarial images generated from this attack and PGD."
>
> We disagree. In Fig. 4 we plot the difference between clean and perturbed image for our method and the baselines. This shows a very significant difference of the adversarial examples found by our approach versus the prior ones, which is then also quantified by the LPIPS metric in Table 1.
>
> > "The attack methods are only compared on CIFAR - Imagenet would be a good addition as datapoint to the experiments."
>
> Not true, we did evaluate our attacks on ImageNet, please refer to Fig 3 and Section 4.1.
>
> > "No mention of the cost of attack, number of steps to reach the adv example and time taken by barrier method optimization."
>
> We did specify the number of steps $T = 10$ in Section 4.2. The cost depends on the chosen transformation. We have instantiated our attack for three transformations: DCT, DWT and Glow. For DCT and DWT, the additional cost (compared to PGD) is negligible since they are fast linear transforms. For the Glow-based attack, the additional cost is linear in the depth of Glow's neural network.
>
> > "For adversarial training, comparison has only been made with PGD. I believe other methods should be used in the comparison."
>
> We have used TRADES (Section 4.2, second paragraph) which remains the standard for adversarial training (despite being introduced in 2019).
>
> > "Again ImageNet not used for adversarial training, would be a good experiment to add."
>
> The cost of a comprehensive adversarial training on ImageNet is beyond the computational resources we have access to. However, we are releasing the code under a GPLv2 license which allows any third-party to replicate the results on other datasets.
>
> > "Overall the experiments do not showcase the effectiveness of method as adversarial attack or for training robust models compared to other state of the art methods."
>
> We respectfully disagree, see Table 2 for evidence.
>
> > "Typo:
> > Eqn 2 : $\mathbb{R}^n$ ->  $\mathbb{R}^q$?"
>
> Indeed, typo corrected.

---

### Meta-Review · Area_Chair_NmaF · 2023-12-15

**Metareview:**

The paper proposes an adversarial attack method to generate adversarial examples that looks similar to the original example, by leveraging image representations instead of constraining Lp distance constraints in the original space. However, reviewers pointed out several weaknesses of the paper as listed below, so we recommend for rejection.

1. The proposed attack has very low successful rate.
2. The claim that the proposed method can be used to improve robustness is not fully supported by experiments.

**Justification For Why Not Higher Score:**

There are several weaknesses pointed out by reviewers, as listed in the meta review.

**Justification For Why Not Lower Score:**

N/A

---

### Decision · Program_Chairs · 2024-01-16

Reject